

# Spectral characteristics of seismic ambient vibrations reveal subglacial hydraulic changes beneath Glacier de la Plaine Morte, Switzerland

Janneke van Ginkel[1,2], Fabian Walter[2,5], Fabian Lindner[3], Miroslav Hallo[1,4], Mathias Huss[2,5], and Donat Fäh[1]

[1]Swiss Seismological Service SED, ETH Zürich, Switzerland
[2]Swiss Federal Institute for Forest, Snow and Landscape Research (WSL), Birmensdorf, Switzerland
[3]Department of Earth and Environmental Sciences, Ludwig-Maximilians-Universität München, Munich, Germany
[4]Disaster Prevention Research Institute (DPRI), Kyoto University, Kyoto, Japan
[5]Laboratory of Hydraulics, Hydrology and Glaciology (VAW), ETH Zürich, Zürich, Switzerland

**Correspondence:** Janneke van Ginkel (janneke.vanginkel@sed.ethz.ch)

**Abstract.**

Glaciers have a complex hydraulic and dynamic behavior that needs to be investigated to improve our understanding of changes in the cryosphere. To tackle this issue, we employ various passive seismic analysis methods on continuous measurements from a temporary seismic array deployed on Glacier de la Plaine Morte in Switzerland. First, we asses the reliability
of ambient noise Horizontal-to-Vertical spectral ratio (HVSR) measurements to the glacier's dynamic environment. The spatiotemporal variations in HVSR curves are predominantly attributed to changing nearby noise conditions influenced by hydraulic, drainage-related tremors, moulin resonances and anthropogenic sources. A careful analysis of the local noise source variations related to glacier dynamic behaviour in order to distinguish between source and medium changes reflected in the HVSR measurements. Only a few hours of HVSR measurements may lead to biases in the interpretation of the HVSR curve.
Despite the influence of these external factors, with long time series of the HVSR measurements, we successfully detect a spatiotemporal trend in HVSR curves. Notably, an HVSR trough emerges following the drainage of Lac des Faverges, an icemarginal lake that rapidly drains, causing water to flow through a channelized system beneath the glacier. This HVSR trough is indicative of a low-seismic velocity layer at the ice-bed interface. Seismic velocity changes derived by interferometry support the presence of a low-velocity layer at the ice-bedrock interface. Inversion and forward modelling reveal a probable thickness
of this low-velocity layer of 10-30 m and a change in S-wave velocity up to 40 %. This layer has a local extend covering an estimated 4.5 to 27 % of the glacier, as indicated by the spatial variations in HVSR trough throughout the array and an independent water volume estimate. The changing seismic velocities are thus a manifestation of temporal water storage at the glacier bed in response to sudden injection of lake water. Our results highlight the value of long time series of HVSR measurements which show variations in the peak/trough structure that reflect hydraulic changes beneath the ice surface.



## 1 Introduction

On the Earth, climate fluctuations are expected to cause significant impacts on glaciers (Kraaijenbrink et al., 2017) and the potential for natural hazards (Ding et al., 2021). The behavior of ice masses is significantly influenced by geological, thermo-dynamic, and hydraulic processes occurring at the basal interface. Nevertheless, the hardly accessible nature of the ice-bedrock

interface makes mapping of its heterogeneity a challenge. Ground penetrating radar (King et al., 2016) and active seismic techniques (Eisen et al., 2010; Luthra et al., 2016) are commonly used for an investigation of physical properties of the subsurface glacier mass. These methods are not suited for repeated surveys within significant time scales capturing hydraulic changes over weeks to months. Moreover, the extent to which these methods retrieve spatio-temporal basal properties that affect ice flow is debated (Kyrke-Smith et al., 2017).

An alternative way to monitor subglacial conditions with high spatio-temporal resolution and relatively low-cost acquisition is by analysing sustained ambient seismic vibrations (Podolskiy and Walter, 2016; Aster and Winberry, 2017). Studies have shown that the seismic power measured on glaciers can serve as a proxy for subglacial discharge (Bartholomaus et al., 2015) and the hydraulic conditions in the subglacial conduits (Gimbert et al., 2016). Furthermore, the glaciohydraulic tremors can also be used to map the water pathways beneath glaciers in space and time (Nanni et al., 2021). In addition to that, seismic in-

terferometry can monitor englacial and subglacial properties such as ice anisotropy and bedrock material by probing subsurface seismic velocities and changes thereof (Sergeant et al., 2020). Zhan (2019) was able to extract transient velocity changes over 12 years related to the surging cycle of Bering Glacier (Alaska), which he attributes to the transition of an efficient channelized to an inefficient distributed drainage system (and vice versa).

In earthquake seismology, the Horizontal-to-Vertical spectral ratio (HVSR) analysis of ambient vibrations (i.e. seismic noise)

is a widely applied technique that is used to determine a site's resonance frequency ($f_0$), as well as its peak amplitude (Nakamura, 1989; Nogoshi and Igarashi, 1971). The HVSR provides information on subsurface properties like shear-wave (S-wave) velocities, layer thickness and site response to incident seismic waves (i.e. Bard et al. (1999); Albarello and Lunedei (2013); Molnar et al. (2018); van Ginkel et al. (2022)). HVSR measurements are purely passive and thus non-invasive for the environment and only require a single station deployment. The HVSR is obtained by calculating the ratio between the Fourier

amplitude spectra of the horizontal and vertical components of a seismic recording (Nakamura, 1989; Bonnefoy-Claudet et al., 2006). HVSR are typically calculated over a range of frequencies giving rise to an "HVSR curve" with characteristic peak-trough structure. The ambient HVSR is dominated by the diffuse wavefield with a significant contribution of Rayleigh waves, which are dispersive seismic surface waves with elliptical particle motion that depends on the subsurface structure (Fäh et al., 2001).

In cryoseismology, the HVSR curves have been used to invert for velocity profiles of the ice or firn (Lévêque et al., 2010; Chaput et al., 2022), to obtain bedrock topography (Yan et al., 2018), and to identify the presence of basal sediments (Picotti et al., 2017). These studies present HVSR curve computations for short time periods (1-12 hours). However, seismic resonances within the soft ice layer and resulting HVSRs are expected to vary with changes in subglacial conditions on diurnal and seasonal time scales. Seismic wave velocities decrease in the presence of drained basal sediments (Picotti et al., 2017), water or air





(Wittlinger and Farra, 2015; Llorens et al., 2020; Guillemot et al., 2024), making them dependent on the geological setting
as well as spatial and temporal variations hydraulic conditions. Köhler and Weidle (2019) found that seasonality in HVSR is
related to the variability of the active permafrost active layer. At the same time, there exist pitfalls of HVSR measurements in
permafrost monitoring as they are influenced by changing noise conditions in addition to freeze-thaw variations within the soil.
Consequently, given varying thermal and hydraulic conditions in cryospheric processes, short period recordings of ambient
seismic vibrations can lead to biases in HVSR curves.

Here we apply the HVSR method to seismic records of Glacier de la Plaine Morte, Europe's largest plateau glacier in
Switzerland's Bernese Alps (Huss et al., 2013). Variations in the peak/trough structure of the HVSR curve reflect hydraulic
changes beneath the ice surface, and we support this finding with additional noise-based seismic analysis. We interpret these
changes in terms of temporary water storage as a response to the subglacial drainage of a nearby ice-marginal lake. Our results
underline the value of passive seismic noise measurement and the HVSR method to monitor the subglacial environment.

## 2   Setting and Data

Glacier de la Plaine Morte (Fig. 1) is the largest Plateau Glacier in the European Alps and located along the border of Switzer-
land's cantons Bern and Valais. Ice flow is limited and minor crevasses are mostly observed on the outlet glacier Rezligletscher
(Huss et al., 2013). From the 5 km wide plateau, the Rezligletscher tongue flows northwards feeding the Trüebbach stream,
which produces proglacial several waterfalls towards the northwest (outside of Fig. 1). A lake, named Lac des Faverges, has
been forming during the melt season at the south-eastern margin of Glacier de la Plaine Morte (Fig. 1b). Since 2011, outbursts
of Lac des Faverges have caused floods in the Simme Valley (Huss et al., 2013). In 2012, a detailed monitoring of lake level
variations, as well as proglacial discharge level was installed for early-warning purposes by the municipality of Lenk and the
engineering company Geopraevent (Fig. 1d,e). The relation of water level to lake volume was inferred from digital elevation
models acquired after each lake drainage. High-resolution lake level records during the drainage allows to determine the lake
discharge dynamics during the drainage events. In addition, there exist daily surface melt estimates based on long-term, sea-
sonal in situ mass balance measurements at a network of sites, extrapolated and temporally downscaled with a distributed
accumulation and melt model (Huss et al., 2021; dataset).

In 2016, Lac des Faverges' drainage initiated through a moulin at the lake margin (green cross in Fig. 1) feeding the sub-
glacial drainage system towards the glacier tongue (Lindner et al., 2020). The drainage began on 27th of August 2016 (day
of year 240) in the evening, lasted approximately six consecutive days and amounted to a volume of 2 million cubic metre of
water (Fig. 1d). In the first six hours of the drainage, the water escaped rapidly with a discharge of 40 m$^3$/s (Fig. 1e). After the
discharge peak, the lake level dropped to the elevation of the moulin inlet, and the connection between the moulin and the lake
was disrupted. Subsequently, the lake established a new link with the moulin through a supraglacial channel, but this slowed
down the drainage process. The drainage of the lake finished in the night from day 246 to 247.

Multiple seismic arrays were deployed on Glacier de la Plaine Morte during the summer of 2016 (see Lindner et al.
(2019, 2020) for deployment and instrument details). In this study, we use the Lennartz 1 Hz shallow borehole array (PM01-



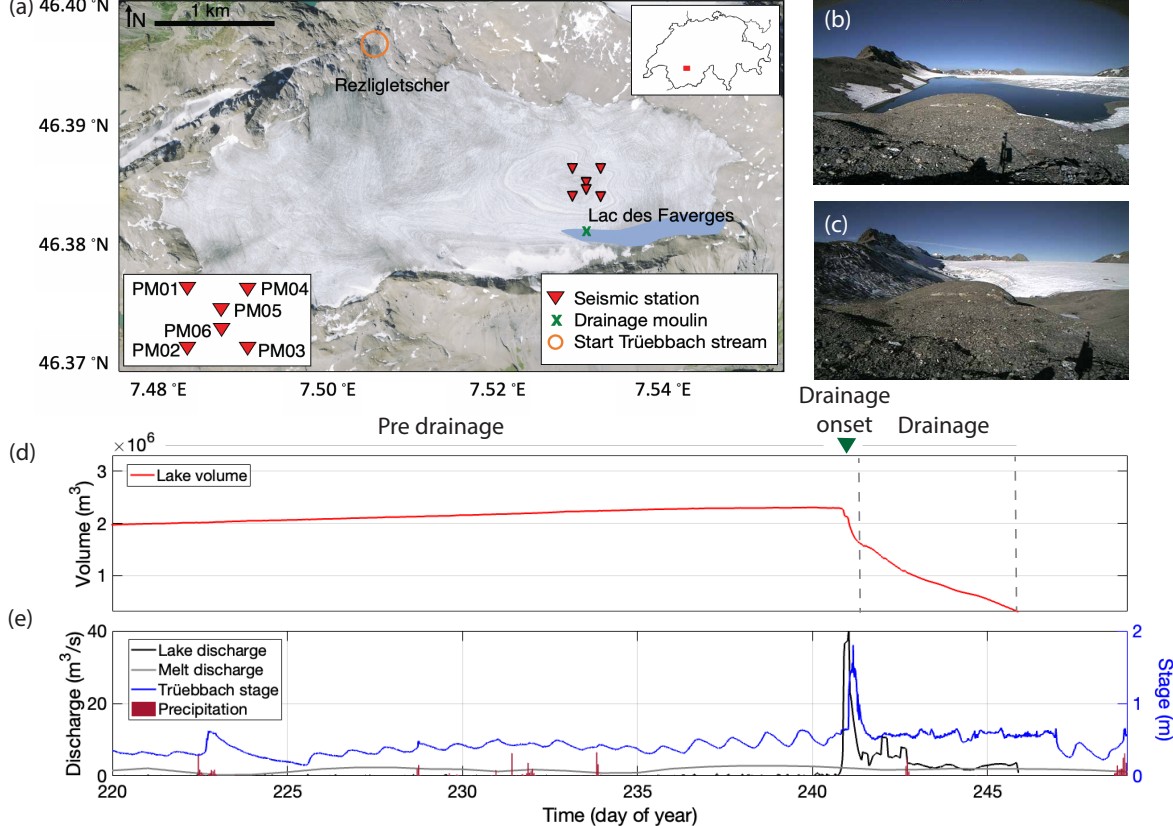

**Figure 1.** a) Orthophoto of Glacier de la Plaine Morte (source: Federal Office of Topography Swisstopo) with the $A_0$ array (stations PM01-PM06, red triangles), the lake (Lac des Faverges) in blue, the drainage moulin (green cross) and the Rezligletscher tongue from which the Trüebbach stream starts (orange circle). The inset in the top-right corner shows the glacier's location within Switzerland. b) Time-laps camera image of Lac des Faverges taken at 27 August 2016 (day 240), 07:00 am, just before the drainage (source Geopraevent, personal communication). c) Time-lapse camera image of Lac des Faverges taken at 06 October 2016, 08:00 am, one month after the drainage (source Geopraevent). d) Lac des Faverges volume curve and the grey dashed lines include the drainage period for which the lake volume and discharge is computed. The pre-drainage and drainage periods are marked, as well as the onset of the drainage (green triangle) e) Lac des Faverges discharge curve (black), modeled mean daily melt-induced discharge (grey) and Trüebbach stage (blue) and precipitation measured at Adelboden (red bars, relative values).

PM06), located 250 m north of Lac des Faverges, deployed from late April (day 120) to early September (day 250) 2016. At the end of July (day 212), a sixth borehole sensor was added to the array (PM06, Fig. 1). We process the three-component seismic
data within the frequency range of 0.5-10 Hz, with a focus on extracting the fundamental seismic resonance frequency of the ice mass using HVSR curves. Over the 4-month recording period, we distinguish a pre-drainage phase (day 120-240) and drainage phase (day 240-248), which is subsequently divided into smaller phases (Fig. 2), depending on the power spectra characteris-



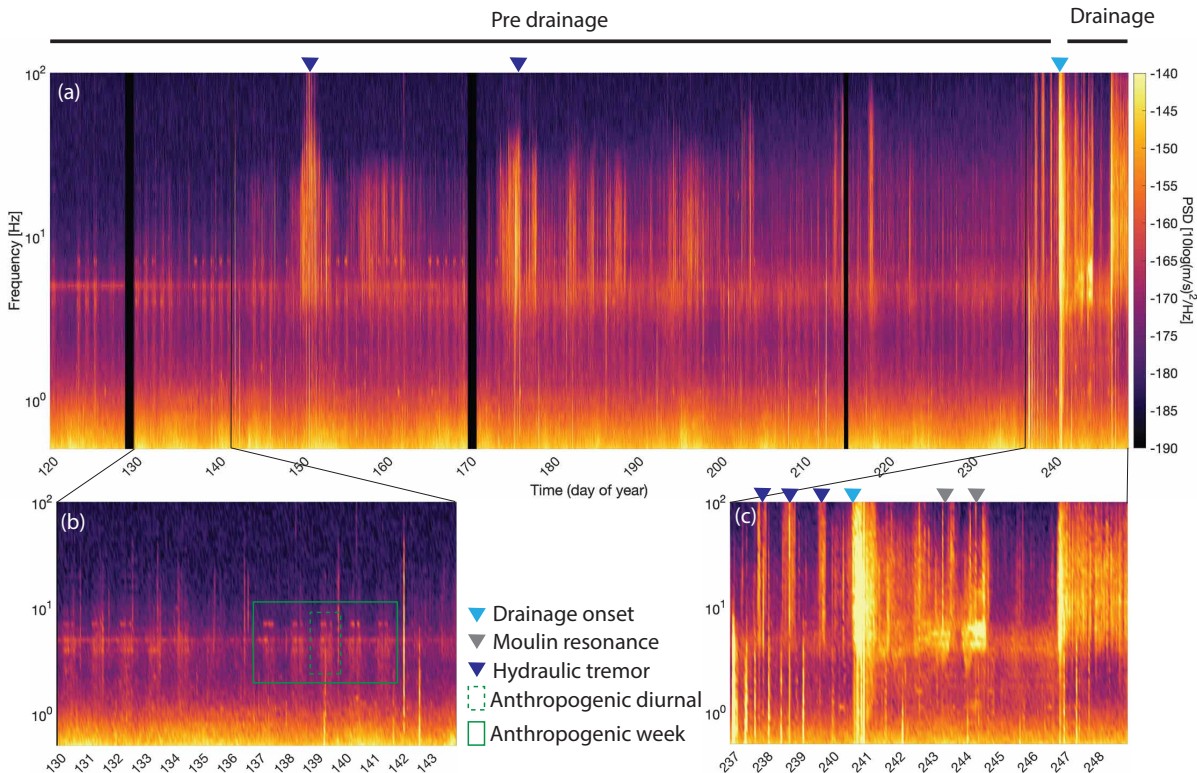

**Figure 2.** a) Spectrogram of station PM02 for the entire recording period. The pre-and post drainage periods are highlighted and the triangles indicate some high-power events like the drainage onset (green), moulin resonance (gray) and hydraulic tremor (blue). b) zoom of the pre-drainage period showing anthropogenic noise. c) Zoom of the drainage period and preceding hydraulic tremor.

tics and knowledge of the subglacial processes from Lindner et al. (2019, 2020). We observe glacial hydraulic tremors during the initial stages of the drainage (day 240). According to Bartholomaus et al. (2015); Gimbert et al. (2016), seismic tremor

in the frequency range of 1-10 Hz are linked to the turbulent water flow and sediment transport within subglacial conduits. In the spectrogram, anthropogenic seismic signals are present at approximately 3-4 Hz and are characterized by a diurnal pattern (power maxima from 06:00 to 18:00 h local time) on weekdays (Fig. 2b).

## 3   HVSR analysis

We compute hourly-averaged power spectra (PSD) for the vertical and horizontal components at all stations in the Glacier de

la Plaine Morte array for days 120-248, following the methodology described in Van Ginkel et al. (2020) to derive the spectral ratios. Ambient noise records of one hour duration are divided into 102.4 seconds long time windows with 75 % overlap





(Peterson et al., 1993), and PSD is calculated for each component within each time window. The PSDs are then averaged over an hour, and the Horizontal-to-Vertical Spectral Ratio (HVSR) is computed using equation 2 in Van Ginkel et al. (2020). As demonstrated by Preiswerk et al. (2019) through polarization analysis, the seismic wave field in Glacier de la Plaine Morte
behaves as in a 1D medium, with no detected 2D or 3D resonances. Hence, Glacier de la Plaine Morte serves as an ideal location for evaluating the potential of the HVSR method.

To characterize the ambient seismic wavefield contributing to the HVSR curves, we employ Matched-Field Processing (MFP, Appendix A) on the Glacier de la Plaine Morte array. The MFP backazimuth within the 3-5 Hz range is selected for presentation in the subsequent sections. This frequency range is chosen because it encompasses frequencies where surface waves become
sensitive to deeper velocity structures, making it suitable for probing the basal interface at a depth of approximately 130-160 m (Grab et al., 2021).

### 3.1 HVSR curves for the pre-drainage period

#### 3.1.1 Resonance frequencies and ice thickness

Between mid-May (day 120) and the end of August (day 237), the HVSR curves are mostly unaffected by strong tremors (Fig.
2). Consequently, all hourly curves within this period are stacked and represented as a probability density function (PDF). From approximately 3100 curves per site, we compute the median HVSR curves (Fig. 3a-f). The stacked HVSR curves at each of the six borehole sites exhibit comparable characteristics: the fundamental resonance frequency ($f_0$) is around 3.0 Hz (with a $\pm$0.5 Hz variability between the stations), and the corresponding HVSR amplitude ($A_0$) is around 1.4 ($\pm$0.2, Tab. 1). During the pre-drainage period, HVSR amplitudes ($A_0$) vary by approximately 20 % (within the 5 % probability) and show a similar
pattern to the power spectra variations (about 8.5 %, Fig. 2), which are attributed to anthropogenic noise with characteristic diurnal and weekly patterns. This argues against melt water or weather conditions being the primary control on noise variations (Appendix B).

Based on the relationship between $f_0$, S-wave velocity ($V_s$) and the thickness ($h$) we can estimate the thickness of the glacier below the stations using (Tsai, 1970):

$$f_0 = V_s/4h \tag{1}$$

For $V_s$ of ice we use the values of 1800$\pm$150 m/s (Podolskiy and Walter, 2016), yielding a range of thickness estimations for the ice. Helicopter-borne ground-penetrating radar (GPR) measured ice thickness of Glacier de la Plaine Morte (Langhammer et al., 2019). Spatial extrapolation of these ice thickness and uncertainties (Grab et al., 2021) are used and compared to the thickness obtained with Equation 1, Table 1. The highest resonance frequencies correspond to the two shallowest GPR esti-
mates. Both GPR and $f_0$-inferred thicknesses for all six stations fall within the associated uncertainties. Variations in thickness among stations within the array likely correspond to local ice thickness variations and glacier thinning towards the east.







**Figure 3.** a-f) Probability density functions from the hourly HVSR curves stacked for days 120-237 (pre-drainage period, in grey box) for the stations PM01-PM05 and for days 212-237 for PM06. g-l) same as (a-f) but for drainage days 245-246, in blue box. The dashed blue circles highlight the trough in PM01 and PM02.

### 3.1.2 Dominant noise sources

Figure 4 depicts the temporal variations in HVSR curves from station PM02 for 97 consecutive days. Generally, HVSR curves up to day 237 demonstrate a stable resonance frequency over time. Exceptions occur during periods of high tremor amplitudes,





**Table 1.** HVSR Fundamental resonance frequency ($f_0$), peak amplitude ($A_0$) from the pre-drainage period, derived from Figure 3. The ice thickness ($h$-GPR) measured by ground penetrating radar comes from Grab et al. (2021), the thickness, $h$-$f_0$, is computed according to Eq. 1.

| Station | $f_0$ median | $A_0$ median | $h$-GPR (m) | $h$-$f_0$ (m) |
|---------|--------------|--------------|-------------|---------------|
| PM01 | 2.99 | 2.19 | 168±13 | 150±12 |
| PM02 | 2.78 | 2.56 | 156±20 | 162±13 |
| PM03 | 3.54 | 2.45 | 129±25 | 127±11 |
| PM04 | 3.17 | 2.47 | 129±14 | 142±12 |
| PM05 | 3.11 | 2.33 | 158±14 | 145±12 |
| PM06 | 2.81 | 2.40 | 140±14 | 160±13 |

where nearby noise sources contaminate the HVSR curves (e.g., day 150 in figure 4). It is important to note that these tremor events, associated with hydrofracturing (Lindner et al., 2020), can persist for hours to days, diminishing the significance of HVSR curves. In Figure 4b, the backazimuth for every hour from day 140 to day 237 for the 2.5-5 Hz band is presented. The relatively low maximum coherence of 0.4-0.5 suggests uncertainty in maximum coherence picks (Fig. A1a,b in Appendix A). Nevertheless, two main noise source backazimuths are identifiable: From day 140 to 182, the dominant noise source exhibits a

backazimuth of around 150°, attributed to anthropogenic noise from roads, industrial facilities, and cities like Sierre and Sion in the Rhone valley to the southeast. From day 182 onwards, the dominant noise source shifts northwestward, towards one of the proglacial waterfalls near the glacier terminus as the melt season progresses. Despite the variability in dominant noise sources, the HVSR curve remains stable during this period.

### 3.2  HVSR curves and noise sources for the drainage period

From day 237 to 248, the seismic power (Fig. 2) exhibits high variations related to the lake drainage. The HVSR time-series from station PM02 (Fig. 5a) is compared against the backazimuth (Fig. 5b) of the dominant noise source and includes these phases related to drainage:

1. Pre-drainage tremor (day 237-240): the strong hydraulic tremor (Fig. 2c) results in the disturbance of the HVSR over the entire frequency band of 0.5-10 Hz. As Lindner et al. (2020) describes, most glaciohydraulic tremors are generated by

subglacial water flow, icequake bursts, or in moulins. The MFP backazimuth results point into the direction of the lake and drainage moulin (at approx. 180°).

2. Onset of the lake drainage (day 240 at 16:00 h): nearby noise source (drainage moulin) interrupts the HVSR curve over the entire frequency band lasting for 8 hours.

3. Ongoing drainage and moulin resonance (days 241-244): ongoing hydraulic tremors cause HVSR disturbances. Moulin

resonances occur in the band of 1.5-4 Hz and generate multiple peaks in the HVSR. This is an example of the effect of a nearby source on the HVSR curves.



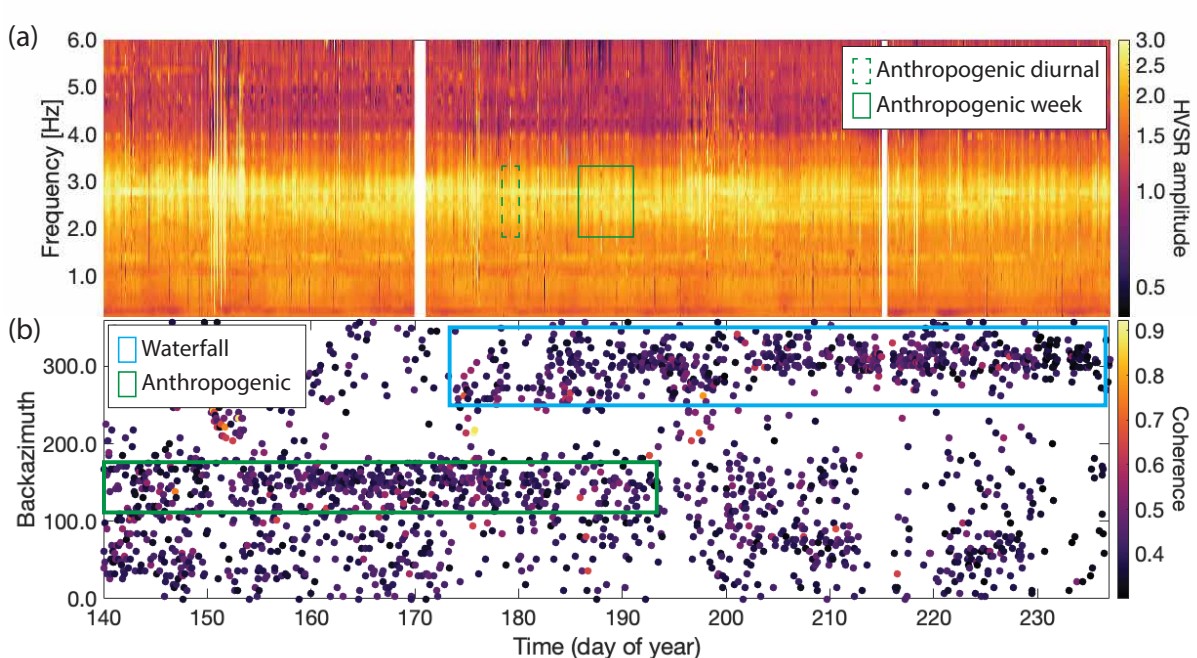

**Figure 4.** a) Time-series of the hourly HVSR curves for days 140-237. The peak amplitudes (around 2.9 Hz) exhibit a diurnal pattern (dashed green box), present on weekdays (solid green box). b) Backazimuth per hour for days 140-237. The cluster around 150° (green box) has an anthropogenic source, and the cluster around 300° (blue box) reflects the waterfalls proglacial water falls of the Trübbach stream (1).

4. HVSR trough (days 242-246): The HVSR curve exhibits a similar peak frequency compared to the pre-drainage phase. Remarkable is the trough that appears at around 4.4 Hz (Fig. 3 g-l and this is visible in Fig. 5a as the dark purple band) and is most pronounced in PM01 and PM02. This trough is also visible at days 242 and 243 during reduced tremor activity. During these periods of trough appearance, the dominant noise backazimuth points towards the proglacial waterfalls in the northwest, with a high signal coherence of up to 0.8. At the beginning of day 242, a trough at lower frequencies appears. However, the high amplitude plateau spanning the fundamental resonance frequency, indicates a disturbance of the HVSR, making the interpretation of this trough unreliable.

5. Days 247-248: Strong tremor and seismic power disturb the HVSR over the entire frequency band. The dominant noise backazimuths of 180-200° points towards the area of the lake drainage.

Based on these observations of the HVSR curve characteristics, we define the appearance of the trough as a primary temporal change most likely related to the glacier's seismic velocity structure. Other changes in HVSR curves coincide with variations of noise source locations and are thus difficult to interpret. In the next sections we present additional seismic analysis to argue that the trough is related to a change in medium properties rather than a varying dominant noise source.



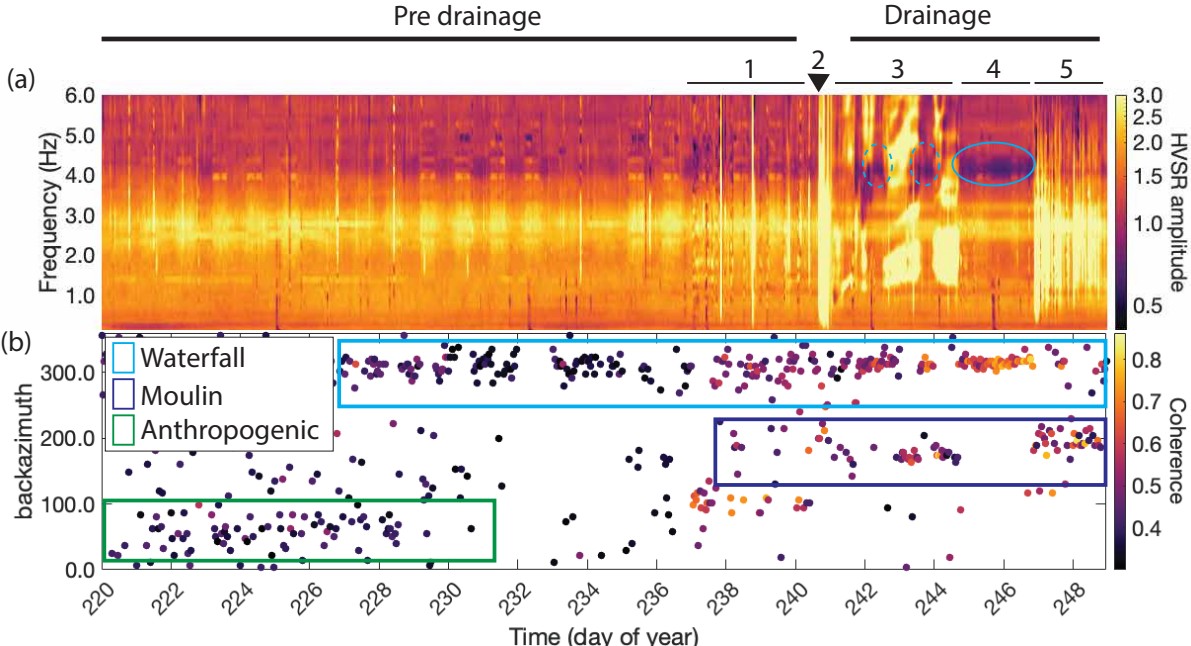

**Figure 5.** a) 2D time-series of all the hourly HVSR curves for station PM02 for day 237-248. The numbers correspond to the numbered list in this section and indicate the different phases related to drainage. The blue dashed circles highlight the trough. b) Backazimuth per hour for the dominant noise source. The green box depicts the cluster for the anthropogenic noise source, the dark blue for moulin and the light blue for the waterfall noise source.

## 4   Seismic interferometry

As additional means of characterizing englacial changes, we employ seismic interferometry. We follow the standard approach to recover virtual seismic surface waves from station pair cross-correlations of noise records (Bensen et al., 2007), and apply it to the phenomenon of a glacial lake drainage (Behm et al., 2020). This may elucidate a varying fracture state and/or water saturation as seismic velocity changes (Zhan, 2019). In temperate glacier applications, the measurement of small seismic velocity changes is challenging as melt-water-induced noise sources are typically non-stationary and scattering of seismic waves is limited (Walter et al., 2015; Preiswerk and Walter, 2018; Sergeant et al., 2020). Here, we make use of the noise source around a backazimuth of 315°, which is stable for several consecutive days and which we identify as a proglacial water fall (Fig. 5b). As this source lies approximately on the axis connecting stations PM01 and PM03, it falls within their stationary-phase zone (Snieder, 2004). The direct wave traveling between the two stations can thus be extracted by cross-correlating the recordings of the water fall noise (Wapenaar et al., 2010; Brenguier et al., 2019, 2020; Mordret et al., 2020).

To this end, we calculate daily stacks of cross-correlations obtained from consecutive 900 s windows of days 230-235 (pre-drainage) and 245-246 (drainage), as these days show a stable waterfall noise (Fig. 5b). For day 246, we discard the last three



hours, since other noise sources become active during that time window. Prior to calculating the cross-correlations, the raw recordings are band pass filtered and subsequently clipped at three times the standard deviation as well as spectrally whitened.

The resulting cross-correlations reside in a narrow frequency band around 4 Hz (Figure 6a and b). As expected, for all considered days, a clear direct wave propagating from PM01 to PM03 emerges, which carries maximum amplitudes near the estimated travel time $t$, obtained by dividing the inter-station distance by a Rayleigh wave velocity of 2800 m/s, see Section 5. Due to the short interstation distance, this direct wave converges towards the auto-correlation, but is slightly shifted to positive lag times, whereas the response at negative lag times is absent because of the one-sided illumination. However, individual

phases exhibit a different travel time between pre and post drainage cross-correlations, and the resulting phase delay $dt$ grows with lag time (Figure 6b). We interpret this delay as a change in seismic velocities and explain the increase $dt$ at greater lag times by a travel time increase of scattered waves, which travel a greater path than direct waves (Snieder et al., 2002).

We systematically investigate this observation by directly calculating travel time delays between the considered days and a reference cross-correlation formed by stacking the six pre-drainage days. For this purpose, we rely on the wavelet cross-

spectrum technique (Mao et al., 2019), which yields $dt$ values as a function of lag time and frequency. To obtain high-quality results, we keep only those $dt$ values associated with a minimum cross-coherence of 0.95 and a minimum cross-spectrum amplitude of 0.9 of its maximum value in the frequency range of 3-5 Hz, which focuses on the region near the estimated travel time. From the remaining $dt$ values, we determine the median and the standard deviation and transform these values to velocity changes $dv/v$ using the relation $dv/v = -dt/t$.

For the pre-drainage period, $dv/v$ values scatter around zero with a maximum deviation of 0.9 % relative to the reference, as shown in Fig. 6c. By contrast, results for the two drainage days show velocity drops between 3 and 4 % (albeit with somewhat higher standard deviation compared to the pre-drainage days), indicative of a medium change. For a detailed depth investigation of the velocity drop, a frequency dependent velocity change would be required, which cannot be obtained due to a less favourable source distribution at higher frequencies (Appendix A). However, compared to a Rayleigh wave traveling within

ice (velocity of around 1800 m/s (Podolskiy and Walter, 2016)), the significantly higher wave speed at above 4 Hz, for which the interferometry results are representative, indicates that the extracted direct wave samples the glacier bed.

## 5   Ice-bedrock velocity model

Next, we use Rayleigh wave ellipticity together with velocity dispersion curves to determine differences in the glacier's S-wave velocity structure between day of year 230 representing pre-drainage conditions and day of year 245 representing drainage con-

ditions. Dispersion curves (Figure 7a) are obtained using high-resolution beamforming (Appendix A) of the vertical component signals. The dispersion curve reveals a plateau and reduced phase velocities for frequencies below 5 Hz during the drainage period. The uncertainties of the dispersion curve below 5 Hz for the drainage period are lower than the velocities for the pre-drainage (Fig. A3),hence we believe these lower velocities are reliable. The ellipticity curves (Fig. 7b) are computed using the RayDec tool developed by Hobiger et al. (2009). RayDec extracts Rayleigh-wave ellipticity from the ambient noise measure-

ments based on the random decrement technique. The shape of the ellipticity curves is similar to the HVSR curves, but the



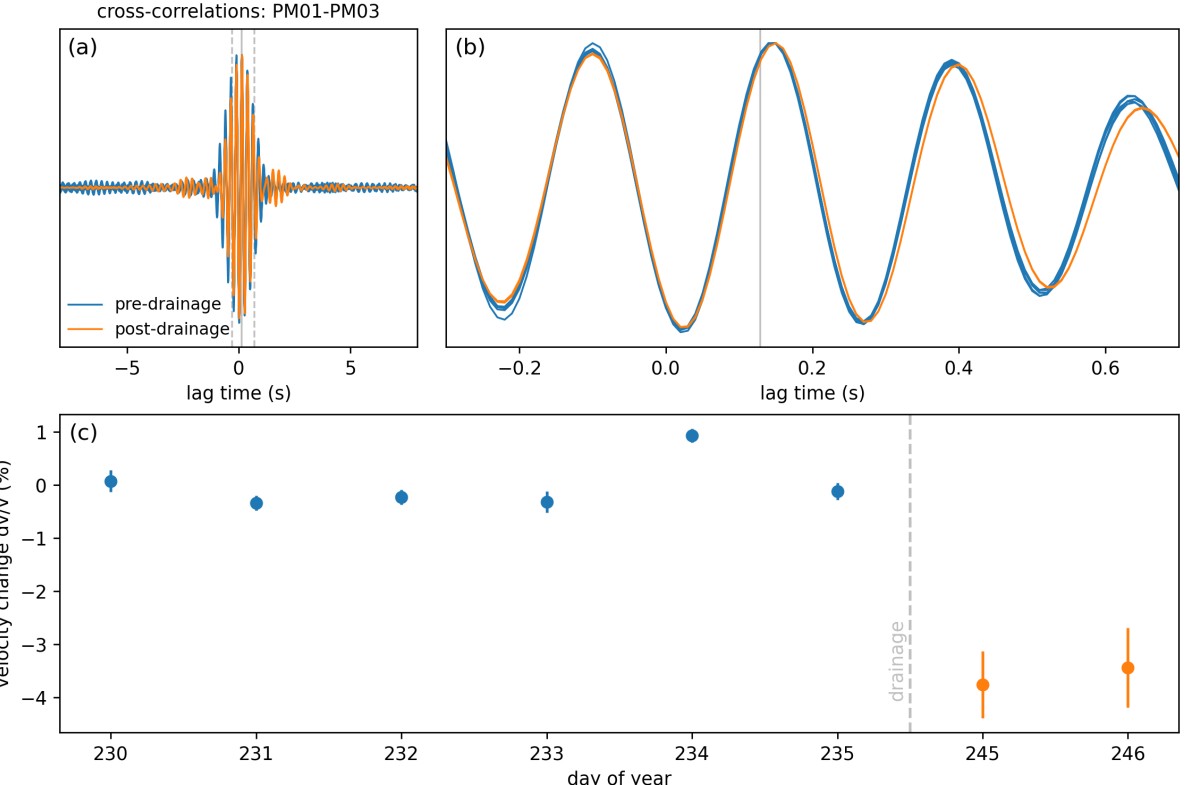

**Figure 6.** (a) Bandpass-filtered (3.5-4.5 Hz) daily cross-correlations between PM01 and PM03 for pre-drainage days 230-235 (blue) and drainage days 245-246 (orange). The solid vertical line indicates the estimated travel time for the direct wave and the dashed vertical lines show the zoom window for (b). (b) Zoom of (a). (c) Velocity change relative to the pre-drainage period. The vertical bars indicate one standard deviation of the $dv/v$ calculation (see text for details).

absolute ellipticity values are smaller than the HVSR measurements, which may include additional signal content from other seismic waves such as Love waves. During the drainage the ellipticity curve exhibits a pronounced trough at about 4.4 Hz, which is a feature not present in the pre-drainage ellipticity data.

### 5.1 Inverse modeling

We perform joint ellipticity and dispersion inverse modeling in the probabilistic framework that includes uncertainties and we supplement it by subsequent forward modeling. We use the inversion algorithm from Hallo et al. (2021) that produces the complete posterior probability density function (posterior PDF) of velocity model parameters. The main advantages are the automatically optimized number of layers of the 1D velocity model and solution uncertainty. The main output of this inversion approach are posterior marginal PDFs of $V_s$, while the P-wave velocities are only supplementary and constrained by an a-225 priory range of Poisson ratio. This inversion includes additional site-specific constraints, like the depth of the bedrock (max.





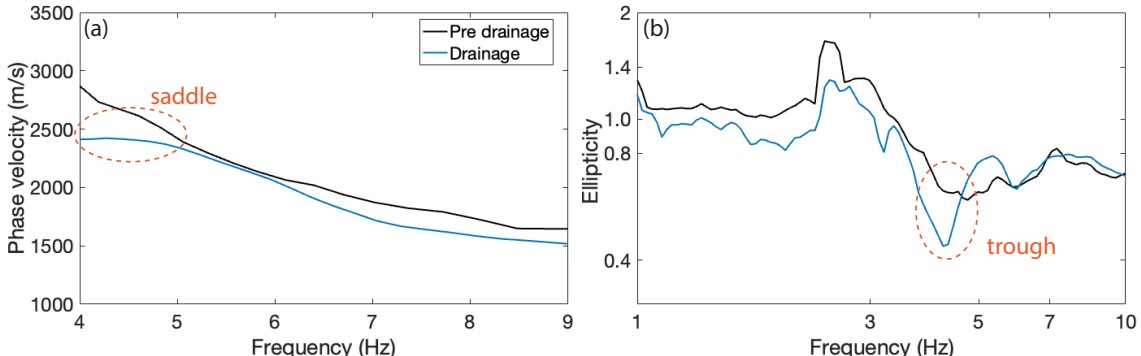

**Figure 7.** a) Dispersion curves for the pre-drainage day (black) and the drainage day (blue), the orange dashed circle indicates the saddle in the drainage dispersion curve. b) Ellipticity curves for the pre-drainage day (black) and the drainage day (blue), the orange dashed circle indicates the trough in the drainage ellipticity curve.

250 m), range of seismic velocities ($V_s$ 200-3500 m/s, $V_p$ 800-7000 m/s), densities (917 kg/m³) and the range in Poisson ratio (0.2-0.49; (Turcotte and Schubert, 2002)) are incorporated.

The results of the inversion and the fitted input data are shown in Figure 8. For day 230 (pre drainage), the data fit for the ellipticity and dispersion curves is satisfactory (Fig. 8c-d). Posterior marginal PDFs of $V_s$ are sharp in the uppermost 130 m
(i.e. the zone of precisely inferred model parameters). In the upper 20 m there is a continuous velocity gradient without any distinctive strong interface, that can be interpreted as the near-surface ice layer subject to weathering and fractures (Lindner et al., 2019; Cook et al., 2016). From 20-130 m depth there is a continuous layer of ice of with expected S-wave velocities (around 1800 m/s). Between a depth of 130-160 m, there is an interface representing the ice-bed transition. The resolution in the probability of $V_s$ is compromised below 200 m as the network is too small to provide dispersion measurements for
frequencies sensitive to depths within the bedrock (grey area).

For day 245 (drainage), two inversions are performed. The first inversion (Fig. 8e-f) is run with a constraint that no low-velocity layer is allowed. This PDF $V_s$ profile is similar to the day 230 profile with an ice-bedrock interface between 130-160 m. However, this inversion run cannot find a velocity model that explains the observed phase velocity dispersion between 4 and 5 Hz (Fig. 8g). The fit of the ellipticity curve also has some discrepancies for the trough at 4.4 Hz (Fig. 8h); hence, a 1D
structure without a low-velocity layer is the wrong seismic velocity model.

The second inversion for day 245 includes a low-velocity layer at the ice-bedrock interface. The fit for the ellipticity is satisfactory but the dispersion curve fit fails to reproduce the saddle point in the lower frequencies (Fig. 8k,l). However, the modeled dispersion curves exhibit a better fit with the measured dispersion than when a low-$V_s$ layer is prohibited in the modeling. The resulting $V_s$ profile shows a shift to lower S-wave velocities at approximately 120 m, which is missing in the
pre-drainage profile. The latter inversion may imply that a low-velocity layer may help to explain the observed ellipticity curve. In particular, in the PDF $V_s$ profile, the upper structure up to 120 m displays a correct S-wave velocity of ice (approximately





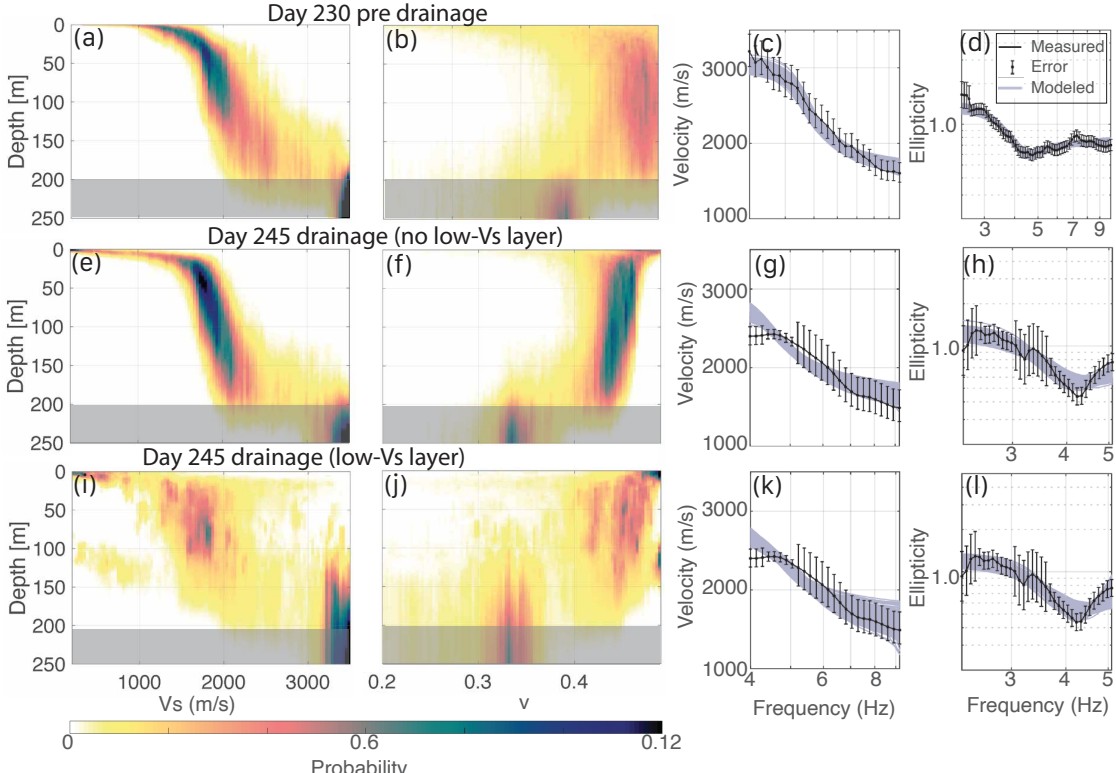

**Figure 8.** Inversion results for a) the pre-drainage period (day 230) a shear-wave velocity profile $V_s$, b) Poisson ratio $v$, c) measured dispersion curve and uncertainty (black line) and modelled dispersion (gray lines), and d) measured ellipticity curve and uncertainty (black line) and modelled ellipticity (gray lines). e-h) inversion results for the drainage (day 245) excluding a no low-velocity layer. i-l) Inversion results for the drainage including a low-velocity layer in the inversion. The inversion results behind the grey shaded region below 200 m (Panels a, b, e, f, i and j) are poorly constrained as a result of lacking dispersion measurements below 3 Hz.

1800 m/s). Between 120-150 m depth, a cluster of probable low-$V_s$ values of 750-1000 m/s appears (Fig. 8i). At the same depth interval, the Poisson ratio ($\nu$) increases to the value of liquid water (0.49) (Fig. 8j). Despite the fit with the low-velocity layer is not optimal, there still is an indication for this layer when compared to the fit of Day 130.

## 5.2 Forward modeling

To further investigate the possibility of the presence of the low-velocity layer, we performed forward modeling in our hypothetical model, which is constrained by the S-wave velocity, depth and thickness of the low-velocity layer. For the forward modeling of the ellipticity and dispersion curves, the Geopsy software package and the modules gpell and gpdc are used (Wathelet et al., 2020). This numerical package can compute dispersion and ellipticity curves of the fundamental mode of Rayleigh waves in

a user-defined layered medium. For the drainage subsurface model, a low-velocity layer is added between the ice and bedrock



with thicknesses of 10-30 m and S-wave velocities ranging between 750-1000 m/s, as inferred from the inversion (Fig. 8i). P-wave velocities remain constant. We compare the model output with the measured dispersion and ellipticity curves (Fig. 9). With a low-velocity layer in the subsurface model, the modeled ellipticity curves display a pronounced trough. Increasing thickness of the low-velocity layer leads to a trough shift to lower frequencies (Fig. 9b). The subsurface models with a velocity

layer of 1000 m/s have the best fit with the measured dispersion and ellipticity.

Revealing the basal low-velocity zone in the inversion process proves challenging due to the lack of low-frequency data in the dispersion curve with the given array setup. Hence, the low-velocity layer probability exhibit significant uncertainty (Fig. 8i). In our particular case, an additional difficulty arises from the conflicting nature of fitting the low frequency of the dispersion curve while simultaneously optimizing the ellipticity curve during inversion. However, from our inverse and forward modeling

tests the low-velocity zone is our favorite explanation of the observed saddle in the dispersion and the ellipticity trough. The low-velocity zone is probably a 10-30 m thick layer with S-wave velocity of around 1000 m/s. Hence, there is a drop of around 40% of S-wave velocity compared to the initial values of 1800 m/s .

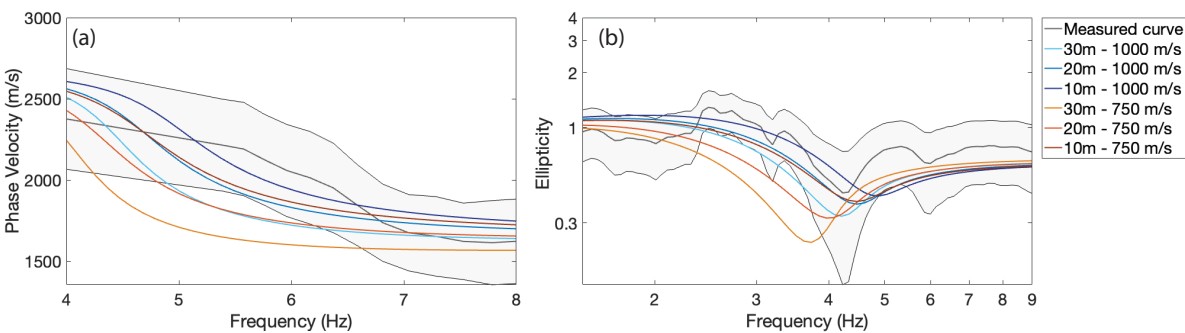

**Figure 9.** Forward modelled a) dispersion curves (red, and blue lines) for several thickness (10-30 m) and velocity (750-1000 m/s) values for the low-$V_s$ layer. The grey line is the measured dispersion curve for day 245 and the grey shaded area the uncertainty of the dispersion. b) Same as for a) but then the measured and modelled ellipticity curves.

## 6    Discussion

Based on the observations from the seismic analyses for the pre-drainage and drainage periods, we propose that during lake

drainage, water is stored englacially in fractures and other void spaces and results in a reduction in the seismic velocity near the base of the ice. In the next sections we discuss how such a transient seismic velocity decrease can be explained in relation to subglacial changes building upon insights from Lindner et al. (2020).

### 6.1    HVSR trough related to medium changes

In our study, we inferred that the HVSR trough adjacent to the fundamental resonance peak (Fig. 3g,h), reflects a change in basal

seismic properties after the lake drainage initiation. Equivalent to our findings, in earthquake site-response studies, a trough in





the HVSR is related to low-velocity layers at depth (Di Giacomo et al., 2005; Panzera et al., 2015, 2019). Hobiger et al. (2021) discuss the presence of trapped surface waves in the low-velocity layer. This resonance influences the vertical component of motion, and deamplification of the horizontal components. Antunes et al. (2022) observed that VHSR peaks (the inverse of HVSR troughs) offer insights into seismic energy anomalies caused by fluids in reservoirs, as the wavefield predominantly

exhibits polarization in the vertical direction. Guillemot et al. (2024) conducted a study using VHSR measurements on Tête-Rousse Glacier in France. They identified subglacial water-filled cavities through the presence of VHSR peaks, a result of diminished basal impedance associated with higher water content.

At Glacier de la Plaine Morte, the trough implies a vertical amplification at around 4.4 Hz. The resonance mode analysis (Appendix C) validates the existence of a vertical resonance around 4.4 Hz for PM01 and PM02, occurring concurrently with

the development of the HVSR trough at these same stations. This implies that the low-velocity layer is spatially constrained or at least spatially varying. Additional evidence for velocity inversions at depth is in a plateau of the dispersion curve, as observed in Figure 7a (Maraschini and Foti, 2010; Chieppa et al., 2020).

The above results are in line with the velocity decrease in the drainage period observed with seismic interferometry. Similar to the HVSR approach, we focused on days with a stable dominant noise source, which lies in the stationary phase zone

of the analyzed station pair. In this situation, a slight backazimuth change of the source (10°), which is not evident in the beamforming results, is unable to produce the observed velocity drop of 3-4 %. As a reference, Zhan (2019) performed noise interferometry across Bering Glacier and reveal 1-2 % seismic velocity reduction. The low-velocity layer can be explained by englacial water storage in basal hydrofractures caused by the drainage, which is expected to drop the seismic velocity of Rayleigh waves. However, the velocity change obtained with interferometry is a bulk measurement representing the entire ice

column. Therefore we cannot make quantitative statements on the velocity drop near the base.

An alternative explanation for the seismic velocity drop involves a sediment layer between the ice and bedrock that becomes water pressurized, a phenomenon identified in Antarctica (Picotti et al., 2017; Guerin et al., 2021). The area around Glacier de la Plaine Morte exposed by rapid glacier retreat during the last years almost exclusively consists of poorly consolidated sediments (Huss et al., 2013). Notably, in our study, we refrained from modeling the presence of a thin sediment layer due to

insufficient information on its thickness and properties. The inferred thickness of the low-velocity layer (section 5) exceeds possible sedimentary layer dimensions.

## 6.2   The subglacial environment

Glacier de la Plaine Morte is a fully temperate glacier and thus similar to Alpine glaciers previously studied with passive seismology (Walter et al., 2008; Nanni et al., 2020). For Glacier de la Plaine Morte, variations in englacial hydraulic conditions

may be the result of varying macroscopic water inclusions, specifically by varying volumes of basal crevasses, void spaces or filling state thereof. The occurrence of basal fracture and void spaces contributes to a decrease in bulk velocity, consequently resulting in a reduction in shear-wave velocity (Biot, 1962). Cracks at the surface of Glacier de la Plaine Morte can lower the Rayleigh-wave velocity by up to 8 % (Lindner et al., 2019). Further observations of anisotropy in glacial ice caused by crystal fabric orientation result in a 3–5 % decrease in shear-wave velocity (Picotti et al., 2015; Smith et al., 2017). At Rhône



Glacier, another temperate Swiss glacier, Gajek et al. (2021) found diurnal variations in shear-wave anisotropy up to 3 %, which they explain by the presence of macroscopic melt water inclusions. The findings presented by Harper et al. (2010) reveal that the subglacial hydraulic system may extend up to 40 m into the overlying ice mass, with hydraulically connected crevasses occupying 0.3 %. To provide a comparison, all englacial fractures (both connected and unconnected) make up 0.46 % of Worthington Glacier, Alaska (Harper and Humphrey, 1995), while all voids, channels, and cracks combined account for

1.3 % of Storglaciären, Sweden (Pohjola, 1994). These numbers for void space and seismic velocity are derived from scenarios involving meltwater-filled fractures. Based on our inverted velocity profiles (section 5), a shear-wave velocity decrease of 40 % is estimated. In the case of Glacier de la Plaine Morte, the occurrence of multi-day pre-drainage hydraulic tremors suggests the establishment of an extensive fracture network (Lindner et al., 2020). Consequently, the quantity of water-filled fractures is likely significantly higher than the numbers presented above, as well as the seismic velocity drop. At Gornergletscher

in Switzerland, a similar ice-marginal lake used to drain annually, releasing a water volume comparable to that of Lac des Faverges (Huss et al., 2007). Here, the water was stored subglacially and englacially and a void ratio ranging from 0.1-10 % was estimated.

### 6.2.1   Hydraulic Parameters

We produced seismic-based evidence for englacial water storage within the ice and in this section we show where this water

originated. For this, we present a comparison of the discharge rate of Lac des Faverges and the stage of the Trüebbach stream (Fig. 1d,e). We include the daily melt-induced discharge of the entire glacier. The daily melt-discharge curve is offset by three days from the Trüebbach stage. To align stage fluctuations for stage-discharge calculations, the melt curve is shifted three days forward. Additionally, the Trüebbach stage peak, delayed by 3.5 hours after the lake discharge peak (Fig. 1e), represents the expected travel time for the subareal part of the draining water between the Retzligletscher tongue and the gauge station.

Consequently, the lake discharge curve is shifted 3.5 hours forward for comparison. The total glacier discharge is then computed by summing the lake discharge and surface melt.

Figure 10 presents the comparison between the Trüebbach stage and the total discharge. The red curves represent the stage-discharge relation (Herschy, 1998) and 95% confidence bounds for the pre drainage period. Here, the stage increases with the discharge. Most meltwater drains northward along the subglacial interface towards the Trüebbach stream. However, a notable

portion of surface melt infiltrates the karstic system connected to the Rhône Valley via subsurface conduits (Finger et al., 2013). This results in a low stage-discharge correlation coefficient ($R^2 = 0.28$). With the onset of the drainage, the discharge rapidly increases initially, followed by an increase in stage. This trend follows the slope of the extrapolated fitting curve of the pre-drainage period (red dashed curve in Fig. 10). However, we acknowledge that the data is below the fit. After the main discharge peak, the orange (from day 241) to black (end of day 245) data points form a cluster where the discharge remains

high while the stage does not surpass the highest values of the pre-drainage stage levels. Over about three days, the discharge more than doubles with respect to pre-drainage conditions whereas the fit of the stage-discharge relation shows no trend in stage. This suggests a significant temporal englacial water storage.





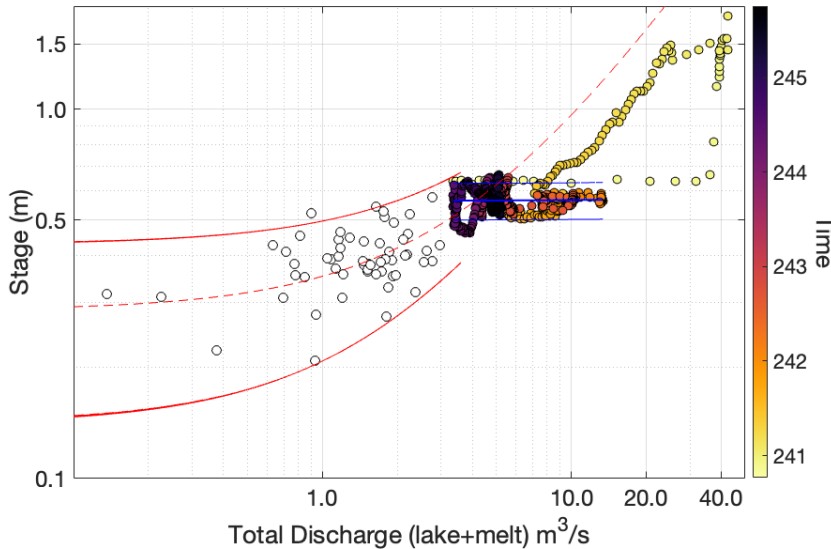

**Figure 10.** Total discharge from the lake and meltwater plotted against the stage of the Trüebbach. The white data points represent the days before the start of the drainage, the color coded data points follow the days from the onset of the drainage until day 246. The red dashed line is the linear fitting function ($R^2 = 0.28$) with the 95 % confidence bounds (thin solid red lines), fitted for the pre-drainage data points. The drainage data points (except the main drainage peak (yellow dots) are fitted in blue.

### 6.2.2 Water storage extent

To obtain an estimation of the aerial extent of the stored water, we assume a low-velocity layer of 10-30 m thick (section 5),
and a void space of 5-10 %. The elevated presence of water-filled fractures implies that the fracture and void volume is likely to be on the high side of the void space numbers provided by Huss et al. (2007). For the discharge volume estimation, the peak discharge volume (Fig. 1e) is excluded, since this water drained directly in the Trüebbach stream. Short glacier surface uplift (Lindner et al. (2020), Figure 2b) implies subglacial water flow for this period. From the lake volume curve, we compute that approximately 1 million $m^3$ water drained from midday 241 up to the end of day 245 into the subglacial system (Fig. 1d). The
total volume of ice filled with water is obtained by dividing the drained water volume by the void space percentage. Given the low-velocity layer thickness, we can then determine the areal extent of the water-filled void spaces. This results in 0.33-2.0 km$^2$ of fractures and voids filled with water. For comparison, the seismic array covers an area of 0.06 km$^2$ and the entire glacier is 7.3 km$^2$ in 2016.

Given the assumption that nearly all the water following the primary drainage peak was temporarily stored in basal fractures,
the impact on the entire glacier area is estimated to range from 4.5 % to 27 %. This value is subject to significant variability due to uncertainties in fracture volume and this estimate serves as an upper limit, as the entire 1 million $m^3$ of water may not be stored simultaneously. The troughs in the HVSRs are primarily observed at stations PM01 and PM02 (Fig. 3), in the western





segment of the array. As depicted in Figure 9 by Lindner et al. (2020), a channelized subglacial drainage system emerges to the south and west of the seismic array and supports the observations that the low-velocity layer has only a local extent.

### 6.3 Implications for HVSR measurement on ice

The HVSR time-series observations on Glacier de la Plaine Morte confirm the general recommendations and known issues of the stability of the HVSR method (Chatelain et al., 2008; Köhler and Weidle, 2019), especially with regards to the impact of varying noise sources. Nevertheless, based on the temporal evolution of the HVSR curves, we are able to identify a medium change at the base of the glacier. Hence, we outline a series of considerations for applying HVSR to a glacier:

1. The HVSR technique is valued for its simplicity as a single-station approach, and computing HVSR curves is straightforward. For ice-thickness estimations relying on resonance frequency, a single-station measurement can be adequate, following the approach outlined in points 2 and 3.

2. Power spectra over the full recording period, covering a wide frequency range can be used to identify events like glaciohydraulic tremors, water flow or moulin resonances. With this transient events in the glacier, the HVSR curves exhibit hourly variations that could introduce bias in thickness or velocity estimations. In scenarios with a nearby seismic noise source, the overall HVSR amplitude tends to rise, accompanied by the appearance of broad HVSR amplitude peaks. In cases of a high event rate, a multiple day to weeks-long recording period is recommended.

3. When the seismic record length is adequate, hourly computed HVSR curves can be stacked and presented as probability density functions (refer to Fig. 3) to achieve a stable estimation of $f_0$. Additionally, one can present the HVSR as time-series to examine the variations in curve characteristics over time.

4. For a comprehensive understanding of spatiotemporal variations and medium changes on such a dynamic subsurface, requires an array of multiple seismic stations. An array facilitates additional passive seismic analyses, including beamforming, seismic interferometry, resonance mode analysis, and phase velocity dispersion, thereby providing valuable insights for site characterization.

5. With an array available, a beamforming analysis allows for acquiring noise source location and phase velocities. The analysis should cover multiple frequency bins, including the one targeting the basal interface. Precise noise source localization is pivotal for distinguishing whether variations in HVSR curves result from a source effect or alterations in the medium. (Sub)glacial processes might produce seismic waves with a frequency content that could resemble $f_0$.

### 7 Conclusions

In this study, we analyzed four months of continuous seismic records on Glacier de la Plaine Morte with the HVSR technique to capture spatiotemporal variations at the basal interface. We aim to asses the reliability of the technique in such a dynamic environment. Subglacial processes change the medium properties for propagating seismic waves, but simultaneously, the source



of the seismic ambient noise field varies. HVSR changes primarily originate from fluctuations in nearby noise conditions influenced by hydraulic tremors, drainage-related events, moulin resonances and anthropogenic sources. Despite these source-related influences, a discernible spatiotemporal trend in HVSR curves reflects material changes within and/or beneath the ice. An HVSR trough emerges after the subglacial drainage of Lac des Faverges, an ice-marginal lake. Seismic velocity changes of 3-4 % inferred from interferometry indicate a velocity drop at the ice-bedrock interface. Inversion and forward modeling of Rayleigh wave ellipticity and dispersion curves further refine the dimensions of this low-velocity zone. A low-velocity layer thickness of 10-30 m is estimated, and a seismic velocity drop up to 40 %. Based on the seismic analysis and discharge anomalies appearing after the drainage, we suggest that the basal low-velocity layer can be explained by englacial water storage in newly formed fractures after drainage. This layer has a local extent, reflected in the spatial variations in HVSR trough throughout the array and covers an estimated 4.5-27 % of the glacier area. Our results thus demonstrate that the HVSR technique can be used as an indicator of a spatiotemporal medium change at the ice-bedrock interface.

We suggest that a deeper understanding of the seismic noise field and the response of HVSR to a dynamic and changing subsurface is needed to make a reliable characterization of the medium using the HVSR seismic method on a glacier. In addition, stacking or filtering multiple day to weeks-long time series of hourly HVSR data is necessary to smooth transient events and changes in noise sources, ensuring a reliable fundamental resonance frequency. Relying solely on a few hours of measurements could result in a significantly biased HVSR curve. The single-station HVSR methodology has demonstrated its potential on glaciers. The next logical step could involve its application in remote and less noisy regions, such as the Antarctic or Arctic ice sheets. Subsequently, a study focusing on temporal variations in HVSR over multiple seasons could be pursued.

*Data availability.* The seismic data used for this research are available in Swiss Seismological Service (Zurich, 1985) at ETH Zurich (http://networks.seismo.ethz.ch/networks/4d/) https://doi.org/10.12686/SED/NETWORKS/4D

## Appendix A: MFP and dispersion curves

Matched-field processing (MFP) is an array processing technique that is based on phase delay measurements. Complementary to the traditional plane wave beamforming, MFP has the advantage that it can also include non-planar waves from of nearby noise sources. MFP identifies the location of a source by a cross-correlation of the observed data with replicas of expected signals computed for various trial source positions. Maximum correlation corresponds to the most likely source location. For further methodology details we refer to Corciulo et al. (2012); Chmiel et al. (2016). Lindner et al. (2020) used the MFP technique to obtain the source locations of the hydraulic tremor at frequencies around 12 Hz. In this study, we used the planar wave assumption to perform a grid search over backazimuth and phase velocity from the ambient vibrations and Figure A1 displays a few examples of the maximum coherence locations. The backazimuth of the dominant noise source are computed for every hour. For the 4-months recording period, backazimuths are computed in three frequency bands: 3-5 Hz, 5-8 Hz, and





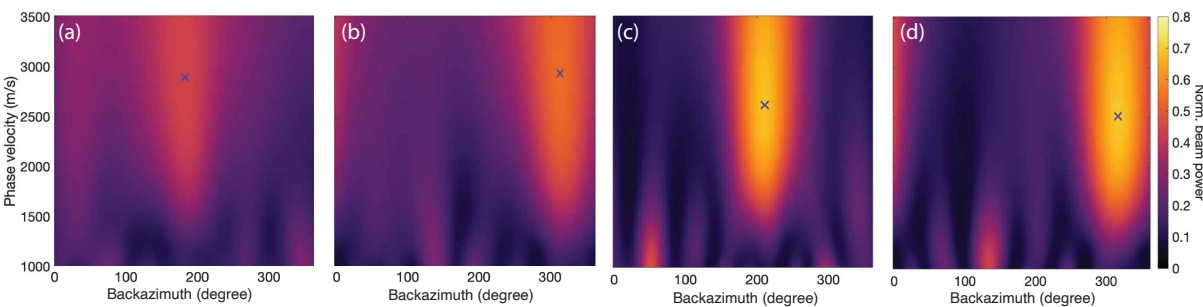

**Figure A1.** a-d) MFP backzimuth and phase velocity for 1 hour of data for resp. day 163, 231, 240 and 246 for the frequency band 2.5-5 Hz. The blue cross identifies the maximum coherence.

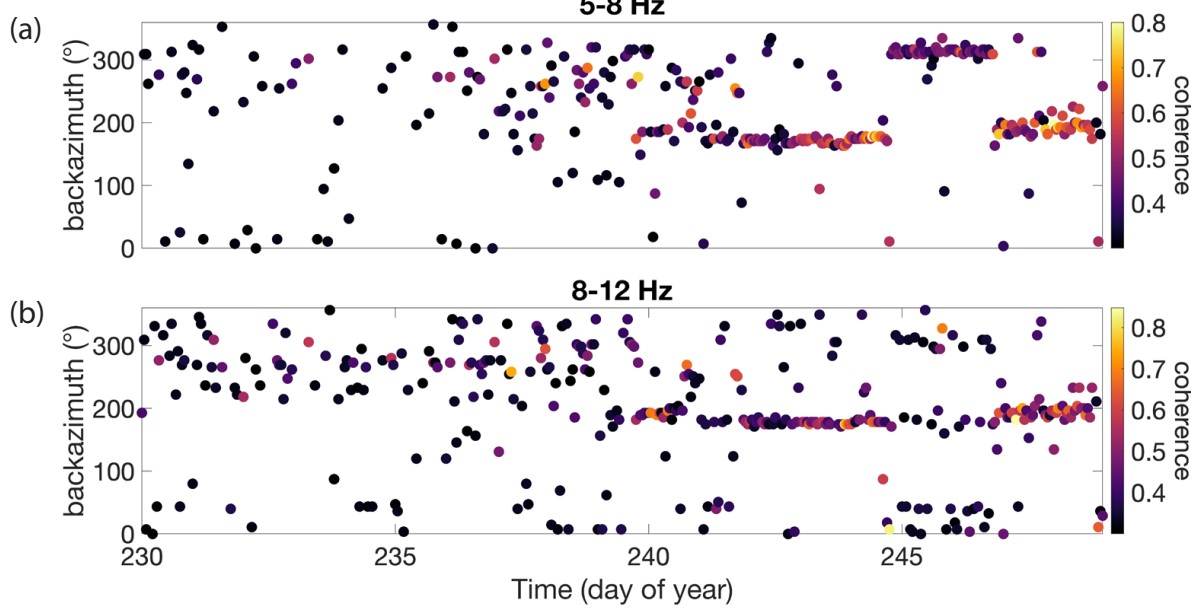

**Figure A2.** MFP backazimuth for days 230-248 computed for frequency band a) 5-8 Hz and b) 8-12 Hz)

8-12 Hz (Figs. 5 and A2). 3 Hz is the minimum frequency regarding the array resolution (Poggi and Fäh, 2010). We found that for this array configuration, a minimum bandwidth of 2 Hz is required to obtain a stable MFP result.

Dispersion curves as presented in Figure 7, are obtained using high-resolution beamforming, the methodology developed by (Poggi and Fäh, 2010) applying the Capon algorithm. The picked mean dispersion curves of probability density functions (Fig. A3) are smoothed, resulting in the dispersion curves of Fig. 7a. We focus on the fundamental Rayleigh wave mode, which dominates the wavefield.



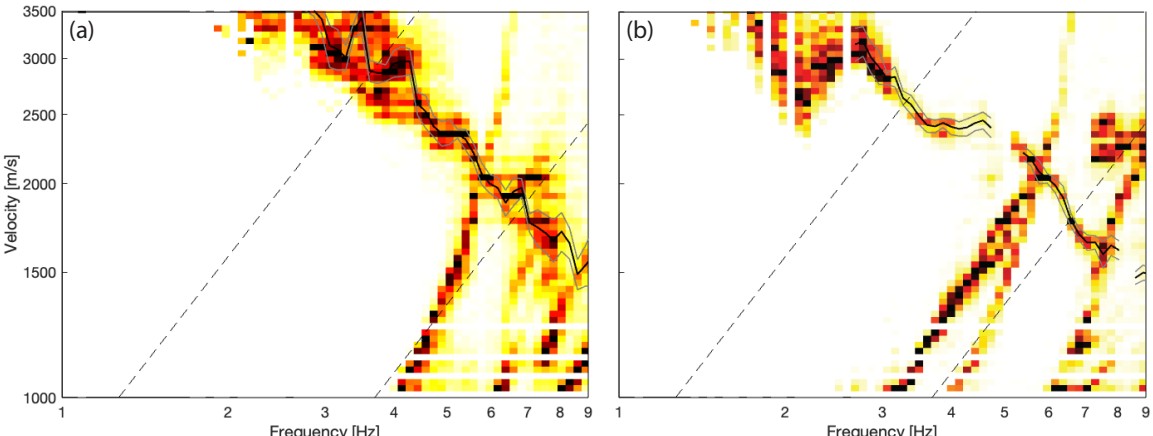

**Figure A3.** Probability density distribution for the dispersion curves obtained by high-resolution beamforming (Poggi and Fäh, 2010). The dashed lines are the array resolution limits, the black line comprises the mean curve and the gray lines the uncertainties for the dispersion curve of pre-drainage day 230 (a) and the drainage day 245 (b). The steep inclined stripes at higher frequencies are aliasing effects.

## Appendix B: HVSR profile and meteorological parameters

Meteorological data are provided by MeteoSwiss weather stations at Montana (1495 m a.s.l., at 8 km from the glacier) and at Adelboden (1320 m a.s.l., 13 km). We use continuous measurements with a 10-minute sampling rate of air temperature, wind speed, solar radiation and precipitation. The HVSR is juxtaposed with meteorological parameters to explore potential correlations, particularly investigating whether the external cause of the diurnal peak amplitude pattern can be identified (refer to Fig. B1). While temperature exhibits a diurnal trend, the intensity of peak amplitude shows no direct correlation with the

fluctuations in peak temperature. Additionally, precipitation, wind, and solar radiation do not exert any discernible influence on the HVSR curves.

## Appendix C: Resonance mode analysis

We employ normal mode analysis of ambient vibrations to retrieve physical parameters that characterize a structure, and to monitor the structure over time. In this analysis we use the frequency domain decomposition (FDD) technique by Brincker

et al. (2001), applied to rock slopes including higher vibration modes (see Section 3 of Häusler et al. (2019)). The FDD output consists of singular values (SVs) and corresponding singular vectors from the singular value decomposition of the cross-power spectral densities of all seismic input traces. The SVs can be understood as the auto spectral densities of the modal coordinates, which peak at fundamental and higher mode resonance frequencies of the structure. The corresponding singular vectors are interpreted as the three-dimensional mode shapes of the structure. The FDD technique is sensitive to varying noise sources, as

it assumes uniformly distributed sources with near-white source spectrum.



**Figure B1.** HVSR time-series of station PM02 for days 120-170 and the meteorological parameters for the nearest weather station ABO (source MeteoSwiss). a) Precipitation, b) Temperature, c) Wind, d) Solar radiation

In this study we performed the mode analysis for 2 hours of the 3-component seismic data applied on four days: the pre-drainage (day 231), drainage event (day 240), moulin resonances (day 242) and the day with the trough in the HVSR (day 245). The noisy bandwidths are included for days 240 and 242 and this nearby moulin noise source disturbs the signal significantly and resonance modes are not identifiable (Fig. C1b,c). Day 231 and 245 (Fig. C1a,d) have the same dominant noise source (one

of the proglacial waterfalls, Figure 5b)), hence we can make a comparison of the pre-and drainage mode analysis. Day 231 displays a fundamental mode ($f_0$) at approx. 3 Hz and a weak second mode ($f_1$) at approx. 4.4 Hz. At day 245, we observe a



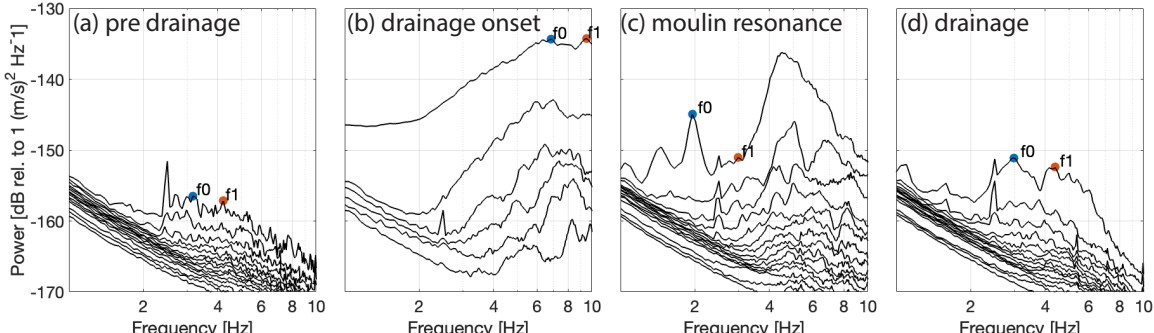

**Figure C1.** Singular values with picked resonance frequencies ($f_0$ and $f_1$) for the a) pre-drainage, b) drainage onset, c) moulin resonance, and d) drainage period.

similar fundamental resonance $f_0$ but the second mode (4.4 Hz) is much more pronounced than at day 231. For $f_1$, the vertical singular vector is dominating, while for the $f_0$, the horizontal singular vectors are strongest.

Table C1 displays the singular vector values. In the resonance mode analysis, particular attention is given to the $f_1$ Z
component (associated with drainage, found in the last column) for PM01 and PM02. Notably, these values are significantly higher than the corresponding horizontal vectors. This distinction between the components is more pronounced here compared to the other stations and the pre-drainage period.

**Table C1.** Singular value vector values for the pre-drainage and drainage resonance mode analysis. $f_0$ is the fundamental resonance mode, f1 the second mode and X,Y Z the mode shape vector of the three components. Since the stations are not geographically oriented we cannot put coordinates to the X,Y vectors and should be considered as relative values. Z is the mode shape of the vertical component.

|  | Station | $f_0$ X | $f_0$ Y | $f_0$ Z | f1 X | f1 Y | f1 Z |
|---|---|---|---|---|---|---|---|
|  | PM01 | 1.0 | -2.14 | 0.97 | 1.0 | -2.17 | 3.6 |
|  | PM02 | -3.2 | -1.1 | 0.93 | 2.54 | -0.68 | 3.26 |
| **Pre drainage** | PM03 | -1.33 | -0.32 | 0.48 | 2.5 | -0.37 | -0.55 |
|  | PM04 | -1.11 | 2.1 | 0.84 | 0.75 | 3.37 | 1.43 |
|  | PM05 | -1.47 | -2.64 | 1.05 | 2.03 | 3.3 | 2.3 |
|  | PM01 | 1.0 | 0.67 | 1.12 | 1.0 | -0.06 | 3.54 |
|  | PM02 | -1.2 | -1.0 | 1.34 | 0.88 | 0.26 | -2.13 |
| **Drainage** | PM03 | 1.39 | 0.32 | -0.7 | 0.73 | -0.42 | -0.48 |
|  | PM04 | -1.43 | -1.41 | -0.8 | 0.89 | 1.59 | 0.91 |
|  | PM05 | 0.95 | -2.16 | -1.31 | 0.42 | 1.29 | 1.83 |



*Author contributions.* JvG initiated the research idea, performed the HVSR analysis, performed the inversion and forward modeling, and wrote the manuscript. FW advised on this project and wrote the manuscript. FL performed the seismic interferometry and wrote the manuscript. MHallo advised on the inverse and forward modeling and wrote the manuscript. MHuss provided lake and melt-discharge data, advised on the glaciology and edited the manuscript. DF advised on the HVSR analysis and edited the manuscript.

*Competing interests.* The authors declare that they have no competing interest.

*Acknowledgements.* The authors thank their home institutions and all people involved in the monitoring systems used in this research. Special thanks go to Mauro Häusler for his contributions to the resonance mode analysis. We acknowledge Malgorzata Chmiel for advising on the MFP analysis and the discussions on the scope of the research. Geopraevent Ltd. (https://www.geopraevent.ch/) and Lenk municipality (https://lenk-simmental.ch) provided the Trüebbach stage and Lac des Faverges time lapse images. The author acknowledges that ChatGPT (OpenAI, GPT-3, http://openai.com) was used for partially improving English writeups.

JvG is funded with an ETH Zürich Postdoctoral Fellowship. FL acknowledges support through the German Research Foundation grant LI3721/2-1. MHallo is supported by the Japan Society for the Promotion of Science Fellowship P23070 and Grant-in-Aid 23KF0149.



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
