# Peer review of "Spectral characteristics of seismic ambient vibrations reveal changes in the subglacial environment of Glacier de la Plaine Morte, Switzerland"

_EGUsphere, 2024_

## Author Comment (AC1)

**Response to RC2**

Dear Florent Gimbert,

Thank you for your very positive remarks and insightful comments on the manuscript. We appreciate the time and effort that you have dedicated to our manuscript. We have discussed your suggestions and summarized the outcome below. The small technical corrections will be also incorporated in the revised manuscript, which will be uploaded in a later stage.

**Major point:**

- Despite the comprehensive effort the authors put in attributing the observed H/V and dispersion curve changes to subsurface hydrology changes, it appears as still unclear to me what these changes correspond to. The joint H/V – surface wave dispersion curve inversions indicate the low velocity layer might be englacial (between 100-150 m deep), but confidence is low and inversions with or without low velocity layer actually both reproduce the saddle in H/V spectra. The forward modelling done with a water layer at the ice-bed interface convincingly reproduces the saddle in the H/V spectra, but does not reproduce the dispersion curves properly.

  Thank you for raising these important points. The development of the trough in the HVSR and the lower velocities of the dispersion curves correspond to a temporal low-velocity layer. After the lake drainage, the drained water is temporarily stored in the subglacial environment, that decreases the seismic velocity of the ice.

  In order to strengthen our statements regarding your issue about the inversion tests, we quantified the improved misfit between inversion tests without or with low-velocity zones. The misfit is quantified by the variance reduction (VR) between synthetic and observed data weighted by reciprocal errors (see error bars in Figure 8). The VR value of 100 % means a perfect fit, the VR value of 0 % means fit on the edge of the data error bars, and the negative values of the VR mean synthetic predictions out of the range of observed data errors. The inversion without the low-velocity zone provides the maximum likelihood and maximum a posteriori models with fits of VR = 41% and VR = 27%, while the inversion with the allowed low-velocity zone provides fits of VR = 75% and VR = 69%. We added these values in the manuscript because it is an objective and clear measure of the improvement in the fit to observed data.

  Still, the empirical dispersion curve (Figure 7a) has clearly lower velocities at lower frequencies, suggesting a velocity drop in the deeper structure. However, it is beyond the resolution power of our data, to model velocity change on the ice-rock interface and in the uppermost permeable rock layer at the same time.

- Although the authors acknowledge this in the text, they do claim in several instances, such, as, most importantly, in the abstract (line 16-17), that the H/V changes reflect changes in water storage at the ice bed interface. I don't think this claim is convincingly enough supported by their observations. Either the authors provide more convincing evidence that this is the case, or they need to soften their statements throughout, e.g. by saying hydrological changes below the glacier surface, either englacial, subglacial or underground. Actually, even the title is a bit

misleading, because it implies that changes occur below the glacier (implicitly at the ice bed interface), while results suggest it might as well be englacial or in the ground.

We agree that we should be clearer on the statements of the storage. Therefore, we added these lines to the introduction: Here, we define the subglacial environment as the area near the ice-bed interface, which can encompass fractures and voids in the glacier sole as well as the top layer of the bedrock.

Hence, also the title is slightly adjusted to: Spectral characteristics of seismic ambient vibrations reveal changes in the subglacial environment of Glacier de la Plaine Morte, Switzerland

The revised manuscript will be adjusted accordingly to our definition of the subglacial environment and to consistent in terminology.

**Minor points:**

- Lines 7-9: I don't understand the sentence.

   Apologies, a word was missing. The new sentence is: A careful analysis of the local noise source variations related to glacier dynamic behaviour is done in order to distinguish between source and medium changes reflected in the HVSR measurements.

- Lines 102: seems like a lot of past H/V analysis involve removing impulsive events such as icequakes from the catalog. The authors do no do this nor mention it. Could you clarify why ?

   You are correct with this observation. On purpose, we included all the data in the analysis since the study aims to test the sensitivity of the HVSR to spatiotemporal changes either in noise sources or medium properties. By stacking hourly HVSR curves as PDFs for long periods of time, the resonance peak is reliable, while additionally we can study the effect of the events on the HVSR.

- Line 114: statement about tremors is hard to connect to Fig 4. Needs more explanation.

   Based on your suggestion this paragraph is adjusted to: Between mid-May (day 120) and the end of August (day 237), the HVSR curves generally maintain a uniform shape. However, during brief episodes of hydraulic tremors, such as on days 151-154, the HVSR curves are disrupted, preventing the formation of any resonance peaks.

- Figure 3: would be nice to show the frequencies outlines in Table 1 in the figure. Also, Table 1 should have a caption.

   That is a good suggestion, hence figure 3 is edited and the resonance frequency and amplitude of Table 1 are highlighted as the dark blue circle. The caption is adjusted accordingly. Table 1 has the caption above the table (default Copernicus format). But in this preprint version it's hard to recognize. I will make sure in the final typesetting it looks clearer.

- Figure 4: would be nice to show temperature timeries on this figure, as a panel on top.

  Good point and we did investigate the relationship with temperature and Appendix B contains the HVSR time series in relation to meteorological parameters like temperature, precipitation, wind and solar radiation. We deliberately put this in the appendices since we cannot find a correlation between the HVSR and the meteorological conditions.

- Line 158: the trough seems to be existing, although of course less pronounced, before the lake drainage. In fact it seems to gradually increase before the lake drainage, and I wonder whether that is significant or not, and related to surface melt ? It would be nice to comment more on this.

  Yes, melt could be an explanation. We have observed this appearance of the trough from day 229 too. Based on the modelled melt discharge or precipitation measurements, there is not a clear relationship. Due to the progress of the melt season, there might be accumulation of melt water causing the appearance of the trough. Though we do not have clear evidence for this. We will add a sentence on this observation in the revised manuscript.

- Line 192: scattered waves ? really ? these waves are probably not much scattered since observed at times quite close to the direct wave.

  We agree that it is not that clear, if the waveform part under discussion are already scattered waves or still the direct wave, which is not well focused in time due to the narrow bandpass filter applied. As we do not use this part for the determination of dv/v anyway (due to low amplitude and coherence and thus quality reasons), we decided to rephrase this paragraph and leave out the discussion on potential coda waves. We hope this avoids confusion.

- Line 199: do you use the stretching method ? please specify.

  As stated in the text, we are using the wavelet cross-spectrum technique, which is similar to the moving-window cross-spectral technique, but with higher resolution along the time axis and the frequency axis. However, to get dv/v estimates, we employ a statistical analysis of determined dt values for the direct wave, as we do not expect a linear relation in this case between dt and t (in contrast to scattered coda waves). We add a sentence in the revised manuscript to make this more clear.

- Line 241: the low velocity layer is hard to see. Maybe you could highlight it by a circle on the figure ? It is actually quite low confidence, thus you might want to specify this in the text.

  Thank you for pointing this out. In Figure 9 we added a circle to highlight the low-velocity layer and added a sentence on the low probability of it in the velocity profile.

- Line 243: I don't see in which ways the dispersion curve predicted with the low velocity layer shows a better fit than without the low velocity layer. The fits appear

as equivalent to me, which makes the argument weak. Either the authors explain this better or remove and acknowledge the poorly constrained nature of inversions.

As we responded to the comment above, in order to strengthen our statements regarding this issue, we quantified the improved misfit between inversion tests without or with low-velocity zones by using the variance reduction (VR). The VR values from the inversion with low-velocity zones (VR = 75% and VR = 69%) are objectively much higher than in the inversion without it (VR = 41% and VR = 27%). This clearly and objectively quantifies the improved misfit. We added these values to the manuscript.

- Figure 8: the inversion find a Poissons ratio of 0.5. Is that realistic ? I would rather think 0.3 is a realistic value, while 0.5 would really correspond only to water ? Isn't there a problem there ? The grey shaded area is also not specified. Does that correspond to bedrock ? known from what, radar?

The Poisson's ratio of ice is approximately 0.32 and of dry rock 0.2-0.35. However, these values can be higher if the profile is impregnated with liquid water. At the same time, Poisson's ratio is only indicative in our inversion due to limited input dispersion curves. Still, the inferred approximate values of the Poisson's ratio are not equal to 0.5 but always below 0.49. For the ice of 0.4-0.49, and for the rock below of 0.3-0.38. These values are on the high side, but more important is the relative and theoretically correct trend, in which rock has a significantly lower Poisson's ratio than the ice with water above it. We have included the statement about the Poisson's ratio in the manuscript. Also, the x-axis of the Poissons ratio in figure 8b was not correct. See below the updated figure 8.

The grey shaded area highlights the depth zone of the inversions which is very uncertain due to the missing low frequency part in the data. This is briefly mentioned in line 235 and the figures caption. In the updated manuscript this is a bit better explained.

[Figure]

*Figure 1: revised figure 8 for the updated manuscript*

- Line 240 : improve wording, we don't fit a day.

  This sentence is adjusted in the revised manuscript.

- Line 265: Convincing from the forward modelling, not as much from the inversion. The authors should acknowledge this, as well as clearly highlighting that the inversion can reproduce the saddle without low velocity layer. They should also explain what makes their inversion able to explain the saddle without accounting for a low velocity layer.

  The forward modelling of the dispersion curves is just a simplistic approximation because of the lack of observed data to fully model the complex wavefield influenced by a hypothetical water layer. Also, formed conduits and the base and the low-velocity layer might act as waveguides. While the modelling is a simple 1D approach. To respond to this comment, we added the statement into the revised manuscript, that the forward modelling of the dispersion curves is just a simplistic approximation because of the lack of observed data to fully model the complex wavefield.

- Figure 10: how about the water being routed underground ? I seem to remember Plaine Morte being a karstic environment, favorable for groundwater drainage that would drain water from the lake elsewhere than the main outlet ?

  That is a valid question. Indeed, Glacier de la Plaine Morte is on karstic bedrock, as described in line 335, leading to a low stage-discharge correlation coefficient for the pre-drainage period. Melt water is partly disappearing through the karst (Huss et al., 2013). We cannot completely exclude that some lake discharge water also enters the karstic system. But based on observation from Lindner et al., 2020 and the development of subglacial conduits due to pressure build up and strong hydraulic tremors, we assume that the majority of the water flows through and below the glacier to the main outlet. This is an important point and we will add this to the discussion section.

We hope we cover your comments and are willing to respond to any further questions and suggestions you may have.

Sincerely,

Janneke van Ginkel, Fabian Walter, Fabian Lindner, Miroslav Hallo, Matthias Huss and Donat Fäh

---

## Author Comment (AC2)

**Response to RC1**

Dear Andreas Köhler,

Thank you for your very positive remarks and insightful comments on the paper. We appreciate the time and effort that you have dedicated to our manuscript. We have discussed your suggestions and summarized the outcome below. The small technical corrections will be also incorporated in the revised manuscript, which will be uploaded in a later stage.

- Line 47: "significant contribution of Rayleigh waves" This is most likely true for glaciers with mostly natural sources I guess but note that Love waves have been shown to have a significant contribution in urban areas with anthropogenic sources. Also, body waves may affect HVSRs at lower frequencies. I suggest to briefly mention noise wavefield composition and its effect on HVSR here.

  Good point about the wavefield, I modified the sentence and added the effect of Love and body waves: "The ambient vibration diffuse wavefield has a significant contribution of Rayleigh waves, which are dispersive seismic surface waves with elliptical particle motion that depends on the subsurface structure (Fäh et al ., 2001). Small contributions of Love-and body waves may include the wavefield and contribute to the peaks in the HVSR curve by amplifying horizontal ground motion (Bonnefoy-Claudet et al., 2006)".

- Line 93-94: Just for clarification: Do you have an idea what controls the hydraulic tremors indicated in Fig 2c, day 237-240 before the drainage?

  Thank you for this sharp point, indeed we should mention the tremor of these days. They are most likely some pre-drainage precursor hydraulic tremors. The pressure is building up in the ice, and fractures and conduits formed to enable the drainage later on (See Lindner et al., 2020 Figure 6). I added a line on this in the manuscript.

- Line 125: It is of course very appropriate to use this well-known formula for a depth estimate. But if I'm not mistaken, this originates from theoretical considerations if SH wave resonance is responsible for the H/V peak. There are studies which showed that the peak frequencies from Rayleigh wave ellipticity peak and SH wave resonance are very close, so this is no issue as such. I mention this because of the wavefield composition mentioned above (Rayleigh waves).

  We agree that the frequency of the H/V peak and the resonance frequency of the site (SH wave resonance) are very close but not exactly equal. Hence, equation (1) is only an approximate relationship. To clarify, we added an explanation to the revised manuscript and a reference to a study with a comparison of these two frequencies (Bonilla et al., 1997).

- Figure 5: It seems the through starts appearing already around day 228. Not very clear though, but could this be due to melt water presence?

  Yes, that could be an explanation. We have observed this earlier appearance of the trough too. Based on the modelled melt discharge or precipitation measurements, there is not a clear relationship. Due to the progress of the melt season, there might

be accumulation of melt water causing the appearance of the trough. Though we do not have clear evidence for this.

- 236 ff: The second inversion seems to result indeed in a slightly better fit, but I'm not sure if I would call this a clear improvement visually. Can you quantify the improved misfit?

  Thank you for this helpful comment, the quantification of the misfit strengthened our statements regarding this issue. In the used inversion method, the misfit is quantified by the variance reduction (VR) between synthetic and observed data weighted by reciprocal errors (see error bars in Figure 8). The VR value of 100 % means a perfect fit, the VR value of 0 % means fit on the edge of the data error bars, and the negative values of the VR mean synthetic predictions out of the range of observed data errors. The inversion without the low-velocity zone provides the maximum likelihood and maximum a posteriori models with fits of VR = 41% and VR = 27%, while the inversion with the allowed low-velocity zone provides fits of VR = 75% and VR = 69%. We added these values in the manuscript because it is an objective and clear measure of the improvement in the fit to observed data.

- Line 261ff: I agree that inversion and modelling results support the low-velocity hypothesis. However, the fits in Fig 9 are not very good. Can you speculate a bit what could be the reason why a more complex model is required in your glaciological setting? Additional layers? 2D/3D effects?

  The geopsy software for the modelling of the dispersion curves is not able to exhibit the plateau we observe in the empirical curves. That creates the biggest misfit. A more complex model not with just two layers over a half-space might improve the models, but we do not have a velocity model of the glacier for this period of time of drainage. We expected to have a uniform layer of ice over the bedrock. The presence of pressurized sediments would also act as a low-velocity layer at the base. In our study, we refrained from modeling the presence of a thin sediment layer due to insufficient information on its thickness and properties. The inferred thickness of the low-velocity layer (section 5) exceeds possible sedimentary layer dimensions.

  The glacier has a more or less sheet-like structure and not a deep incised valley. That minimizes the 2D/3D effect. However, formed conduits and the base and the low-velocity layer might act as waveguides. While the modelling is a 1D approach.

  Hence, the forward modeling test is a simplistic approximation, because of the lack of observed data to fully model the complex wavefield influenced by a hypothetical water layer. We added this statement to the revised manuscript.

- Line 296-298: Could then pressurised sediments be an explanation for the misfit? See my comment above.

  In our opinion, the misfit is due to that we are on the low-frequency resolution limit. But I cannot completely rule out pressurized sediments and we add this statement to the manuscript. In the introduction we added this sentence: Here, we define the subglacial environment as the area near the ice-bed interface, which can encompass fractures and voids in the glacier sole as well as the top layer of the bedrock.

This includes the ice-bedrock interface including the bottom part of the ice and the top part of the bedrock, where also maybe some sediments are present. The resolution of the inversion is not sufficient to distinguish in more detail.

Also, Glacier de la Plaine Morte is on a karstic bedrock. Meltwater from the glacier is also disappearing partly via this system and not only via the main outlet in the north We cannot completely exclude that some lake discharge water also enters the karstic system. But based on observation from Lindner et al., 2020 and the development of subglacial conduits due to pressure build up and strong hydraulic tremors, we assume that the majority of the water flows through and below the glacier to the main outlet. This is an important point and will add this to the discussion section.

We hope we cover your comments and are willing to respond to any further questions and suggestions you may have.

Sincerely,

Janneke van Ginkel, Fabian Walter, Fabian Lindner, Miroslav Hallo, Matthias Huss and Donat Fäh

---

## Author Response (AR3)

Dear Kristin Poinar and Andreas Köhler,

Thank you for your feedback on the paper. We appreciate the time and effort that you have dedicated to our manuscript. Please find below our response to your suggestions and updated figures as you requested. The revised manuscript contains the updated figures 8 and 9, as well as some minor text edits. In section 5.3 of the revised manuscript we discuss the preferred the low-velocity scenario.

We hope you appreciate our reply and updated figures and manuscript and we can continue the publication process.

Sincerely,

Janneke van Ginkel, Fabian Walter, Fabian Lindner, Miroslav Hallo, Matthias Huss and Donat Fäh

Response to the suggestions:

Figure 8: There seems to be a mistake here. The measured dispersion curve in (g) is apparently the same as in (c). But the dispersion curve in (g) is supposed to be the one measured during the drainage, i.e. should be the same as in (k) if I'm not mistaken. I'm wondering if this is just a plotting error or an actual wrongly done inversion. This needs to be checked by the authors and may have consequences for the next point.

Thank you for pointing this out. Indeed, there were a mistake in the latest version of the figure, we sincerely apologise for this issue. Nevertheless, that was just visualisation error caused by the figure update after previous revision, not to be interpreted as an error in our analysis. As was also correctly shown in the earlier versions of the paper, the dispersion curve in (g) is not the same as in (c) but as (k). The figure is updated accordingly:

[Figure]

*Updated figure 8*

As to the significance of obtaining a better fit with a low velocity layer, I am also still not 100% convinced. I agree that variance reduction indicates a better fit even through visually there is not clear difference between 8(h) and 8(l). However, I cannot follow the argument that the modelling in Fig 9 supports the low-velocity model better. I would strongly suggest to add the forward-modeled dispersion curve and ellipticity for the no-low-velocity layer case into Figure 9. Only this would allow to decide against or in favor of the low-vel layer. This is necessary because it seems from Fig 8h, that the non-low-velocity layer model is also able to produce a sharp through in the ellipticity.

Thank you for your feedback and we acknowledge your point that Fig. 8h also produces a trough as we use the ellipticity with a trough as input, resulting in a trough in the inversion fitting (this is explicitly stated in lines 256-262 in the revised manuscript). Nevertheless, the trough produced by the inversion without low-velocity zone is weaker compared to the inversion with the low-velocity layer. It is reflected in the mathematically rigorous measure of the data fit by Variance Reduction (VR). Note that there an improvement in the VR in the inversion using a low-velocity layer (lines 257 and 270).

Next, following reviewers recommendation, please find below the update figure 9, where the measured pre-drainage (no low-velocity layer) dispersion and ellipticity and the forward modelled no low-velocity layer is added (green dotted line). Here we do not see such a sharp trough in the ellipticity model as for the drainage measured and modelled ellipticity. This, once again, supports our statements about low-velocity layer, as we discuss in lines 288-292 in the manuscript. Also, in the manuscript we acknowledge that the fit of the dispersion curve is not as good as the ellipticity (explicitly stated in lines 284-285 in the revised manuscript).

The inversion and the forward modeling are two separate tests (independent of each other). As we are aware of the low (blurry) probability of the low-velocity cluster (Fig 8i), we subsequently perform the forward modeling with the constraints on the input thickness and velocities from the inversions. The fact that forward modeling independently implies the same conclusions as the inversion is encouraging. Florent Gimbert, the other reviewer suggested the following: "Convincing from the forward modelling, not as much from the inversion. The authors should acknowledge this". That's why we phrased line 301-302 in the manuscript like this: "The low-velocity layer, particularly highlighted by forward modelling, helps to explain the observed trough in the ellipticity curve, although the saddle in the dispersion curve was not accurately captured"

[Figure]

*Updated figure 9*